# A phase I/IIa safety and efficacy trial of intratympanic gamma-secretase inhibitor as a regenerative drug treatment for sensorineural hearing loss

Anne G. M. Schilder[1,2,3], Stephan Wolpert [4] ✉, Shakeel Saeed[1,2,3], Leonie M. Middelink[5], Albert S. B. Edge [6], Helen Blackshaw[1,2], REGAIN Consortium*, Kostas Pastiadis[7] & Athanasios G. Bibas [7]

Inhibition of Notch signalling with a gamma-secretase inhibitor (GSI) induces mammalian hair cell regeneration and partial hearing restoration. In this proof-of-concept Phase I/IIa multiple-ascending dose open-label trial (ISRCTN59733689), adults with mild-moderate sensorineural hearing loss received 3 intratympanic injections of GSI LY3056480, in 1 ear over 2 weeks. Phase I primary outcome was safety and tolerability. Phase IIa primary outcome was change from baseline to 12 weeks in average pure-tone air conduction threshold across 2,4,8 kHz. Secondary outcomes included this outcome at 6 weeks and change from baseline to 6 and 12 weeks in pure-tone thresholds at individual frequencies, speech reception thresholds (SRTs), Distortion Product Otoacoustic Emissions (DPOAE) amplitudes, Signal to Noise Ratios (SNRs) and distribution of categories normal, present-abnormal, absent and Hearing Handicap Inventory for Adults/Elderly (HHIA/E). In Phase I ($N = 15$, 1 site) there were no severe nor serious adverse events. In Phase IIa ($N = 44$, 3 sites) the average pure-tone threshold across 2,4,8 kHz did not change from baseline to 6 and 12 weeks (estimated change −0.87 dB; 95% CI −2.37 to 0.63; $P = 0.252$ and −0.46 dB; 95% CI −1.94 to 1.03; $P = 0.545$, respectively), nor did the means of secondary measures. DPOAE amplitudes, SNRs and distribution of categories did not change from baseline to 6 and 12 weeks, nor did SRTs and HHIA/E scores. Intratympanic delivery of LY3056480 is safe and well-tolerated; the trial's primary endpoint was not met.

Hearing loss is the most common sensory disorder in humans and an area of significant unmet clinical need[1,2]. The most common cause of hearing loss is age-related progressive loss of inner ear sensory hair cells and/or their synapse[3,4]. Because in humans these hair cells do not naturally regenerate, hearing loss progresses with age. Current treatment of choice with hearing devices focuses on sound amplification; their benefits in understanding speech in noisy environments are limited, therefore many people choose not to use them[5,6]. Importantly, they do not treat the cause of hearing loss. This is where recent discoveries in the molecular pathways leading to hair cell loss and

A full list of affiliations appears at the end of the paper. *A list of authors and their affiliations appears at the end of the paper.
✉e-mail: stephan.wolpert@med.uni-tuebingen.de

regeneration have opened avenues for novel approaches to the treatment of hearing loss. They have allowed for the identification of therapeutic targets and the development of small molecule drugs that may promote hair cell regeneration[7–14].

In young adult mice acutely exposed to noise at a level sufficient to induce loss of hair cells, pharmacological inhibition of Notch signalling with a gamma secretase inhibitor (GSI) upregulated Atoh1, which encodes a bHLH transcription factor required for hair cell differentiation. This approach regenerated hair cells through trans-differentiation of supporting cells and partially restored hearing[14]. Likewise, in an adult guinea pig model of hearing loss due to noise exposure, siRNA silencing of Hes1, an effector of the Notch pathway, caused an induction of new hair cells and partial recovery of hearing[15].

Following the identification and successful preclinical development of a GSI (LY3056480) with an optimal profile[16], we designed and delivered a Phase I and IIa clinical trial. Here, we report on the safety and efficacy of this drug, administered intraympanically in adults with mild to moderate sensorineural hearing loss (SNHL).

## Results

### Characteristics of the patients

From January 24 to October 17, 2018, 15 patients with mild to moderate hearing loss were enroled in the Phase I trial at the UK site (Fig. 1); 3 patients received 3 intratympanic doses of 25 µg LY3056480, 6 received 3 doses of 125 µg, and 3 and 3 patients received 3 doses of 200 µg and 250 µg, respectively. From January 30 to August 5, 2019, 44 patients were enrolled in the Phase IIa trial, 24 at the UK site and 12 and 8 at the German and Greek site, respectively (see patient characteristics in Table 1). All received 3 intratympanic doses of 250 µg LY3056480, except 1 patient receiving the diluent-only at the last injection due to a procedural error.

### Safety outcomes

All Phase I patients experienced one or more AEs. In total 174 AEs were reported of which 2 were probably related to the investigational medicinal product (IMP) and 52 probably or definitely related to the procedure. All AEs with a relation to the IMP and/or procedure resolved. No SAEs were reported.

In Phase IIa, all patients experienced one or more AEs. Of the AEs reported (see Table 2), 277 (80%) were classified as mild, 70 (20%) as moderate and none as severe. The two reported SAEs were CTCAE Grade 2 (moderate) and considered unrelated to the IMP.

Patients reported fluctuations in tinnitus severity across time-points (recorded as AE, tinnitus experience, TFI score). Tympanometry readings did not change from baseline to 12 weeks; all intratympanic injection sites closed. See Supplementary Information for all secondary safety outcomes.

### Primary efficacy outcome

For the total population of 44 patients with mild to moderate SNHL, the primary endpoint was not met, i.e., the mean pure-tone air-conduction threshold across 2, 4, and 8 kHz did not change from baseline to 12 weeks in the treated ear (Fig. 2) (estimated change −0.46 dB; 95% confidence interval [CI] −1.94 to 1.03; P = 0.545).

### Secondary efficacy outcomes

The mean pure-tone air-conduction threshold across 2, 4, and 8 kHz did not change from baseline to 6 weeks in the treated ear (estimated change −0.87 dB; 95% CI −2.37 to 0.63; P = 0.252; see Fig. 2).

Nineteen patients (N = 42; 45%) showed an improvement of ≥10 dB (see methods) in one or more individual frequencies (including 12.5 and 16 kHz) from baseline to both 6 and 12 weeks.

Seven patients (N = 42; 16.7%) showed an improvement of ≥10 dB in two or more adjacent frequencies, or of ≥20 dB in one or more individual frequencies from baseline to both 6 and 12 weeks (see Fig. 3). See Supplementary Information for post-hoc analyses of pure-tone audiometry outcomes at 6 and 12 months.

### Secondary efficacy outcomes

The mean speech reception threshold in noise, measured with a words-in-babble test and expressed as the SRT50n, defined by the signal-to-noise ratio (SNR) that yields an average response of 50% correctly recognised words, did not change from baseline to 6 weeks (estimated change 0.90 dB; 95% CI −0.09 to +1.88; P = 0.074) and 12 weeks (estimated change 0.08 dB; 95% CI −1.01 to + 1.18; P = 0.881) in the treated ear. See Supplementary Information for post-hoc analyses of speech perception in noise measures at 6 and 12 months.

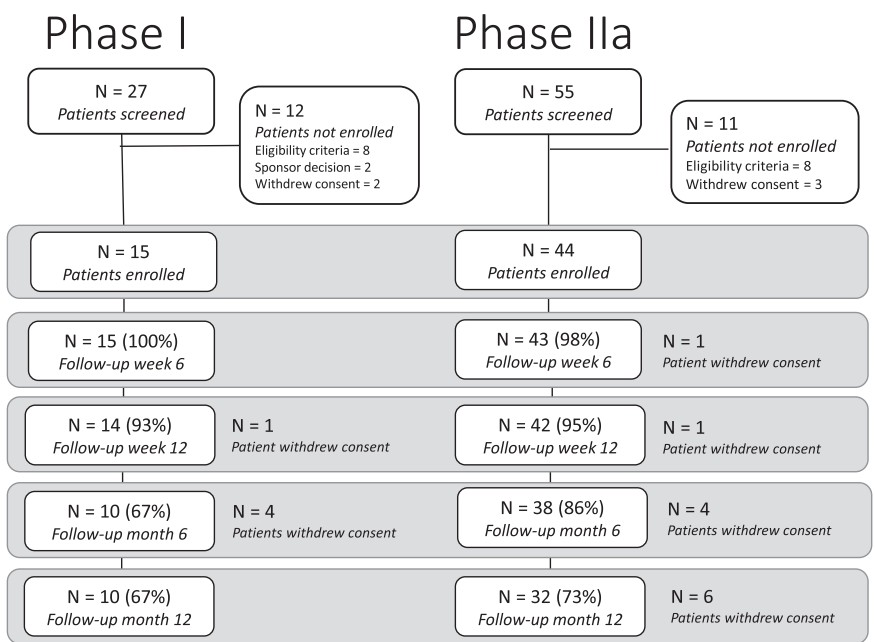

Fig. 1 | Consort diagram phase I and IIa trial. Patient disposition of phase I trial and phase IIa trial.

**Table 1 | Baseline characteristics of trial patients**

| | | Phase I | Phase IIa |
|---|---|---|---|
| Dose level | µg | 25, 125, 200, 250 | 250 |
| Number of participants | N | 15 | 44 |
| Age | Median (years) | 60 | 58 |
| | Min - Max (years) | 22–79 | 32–73 |
| Gender | Female | 7 (46.7%) | 15 (34.1%) |
| | Male | 8 (53.3%) | 29 (65.9%) |
| Years of schooling | Median (years) | 16 | 16 |
| | Min - Max (years) | 12–20 | 8–24 |
| Education | Primary schooling only | 0 (0%) | 1 (2.3%) |
| | Secondary schooling | 5 (33.3%) | 20 (45.5%) |
| | Tertiary / higher education | 10 (66.7%) | 23 (52.3%) |
| Severity of hearing loss | Mild | 9 (60%) | 14 (33.3%) |
| | Moderate | 6 (40%) | 29 (66.7%) |
| | Mean Pure-Tone HLA 2,4,8 kHz | 49.1 ± 12.6 HL | 55.3 ± 9.3 HL |
| Duration of hearing loss | Range (yrs) | 0–10 | 0–19 |

The mean DPOAE amplitude and signal-to-noise ratio (SNR) did not change from baseline to 6 and 12 weeks. Neither did the distribution of DPOAE categories 'present-normal, present-abnormal, and absent' change from baseline to 6 and 12 weeks (Bonferoni correction was used for multiple comparisons). See Supplementary Information for post-hoc analyses of DPOAE outcomes at various time points.

Hearing Handicap inventory of Adults/Elderly (HHIA/E) scores did not change from baseline to 6 weeks ($P = 0.845$) and 12 weeks ($P = 0.51$).

The mean pure-tone air-conduction threshold across 2, 4, and 8 kHz did not change in the untreated ear from baseline to 6 weeks (estimated change −0.84 dB; 95% CI −2.34 to 0.65; $P = 0.265$) and to 12 weeks (estimated change −0.86 dB; 95% CI −2.34 to 0.63; $P = 0.253$). Twenty-one patients ($N = 42$; 50%) showed a pure-tone hearing threshold improvement of ≥10 dB in one or more frequencies in the untreated ear at 6 weeks and 22 patients ($N = 42$; 52%) at 12 weeks. In 24 patients ($N = 42$; 57%) the difference in change from baseline to 6 weeks in pure-tone hearing threshold across 2, 4, and 8 kHz between the treated and untreated ear was ≥10 dB in one or more frequencies, and in 23 patients ($N = 42$; 55%) at 12 weeks, respectively.

The mean SRT50n did not change in the untreated ear from baseline to 6 weeks (estimated change 0.03 dB; 95% CI −0.95 to 1.02; $P = 0.944$) and 12 weeks (estimated change −0.17 dB; 95% CI −1.27 to 0.93; $P = 0.759$).

## Discussion

This Phase I/IIa trial, developed and delivered by our EU academic-industry consortium, shows that LY3056480 delivered intratympanically in adults with mild-moderate SNHL is safe and well tolerated. Our aim was to test the hypothesis that local administration with the GSI LY3056480 restores outer hair cell function and thereby improves the perception of speech-in-noise, which is the primary unsolved problem for people with hearing loss[17,18]. With our trial being the first of a regenerative hearing drug in 2016, we chose a broad set of outcome measures: pure-tone audiometry up to 16 kHz (extended higher frequencies considered key to speech perception)[19,20], speech-in-noise tests as the outcome best reflecting patient experience; and otoacoustic emissions as an objective measure of outer hair cell function[21].

**Table 2 | Adverse drug reactions in participants receiving LY3056480 for SNHL**

| System Organ Class (term) | Event | LY3056480 N = 59 n (%) |
|---|---|---|
| Cardiac disorders | Sinus bradycardia | 1 (2%) |
| Ear and labyrinth disorders | Ear discomfort | 14 (24%) |
| | Ear pain | 13 (22%) |
| | Hypoacusis | 16 (27%) |
| | Dysacusis | 1 (2%) |
| | Tinnitus | 27 (46%) |
| | Ear congestion | 12 (20%) |
| | External ear pain | 1 (2%) |
| | Ear pruritus | 1 (2%) |
| | Vertigo | 1 (2%) |
| Gastrointestinal disorders | Nausea | 2 (3%) |
| | Hypoaesthesia oral | 3 (5%) |
| | Dry mouth | 1 (2%) |
| | Tongue dry | 1 (2%) |
| | Abdominal discomfort | 2 (3%) |
| | Oral pruritus | 2 (3%) |
| | Noninfective gingivitis | 1 (2%) |
| General disorders and administration site conditions | Injection site pain | 49 (83%) |
| | Injection site pruritus | 3 (5%) |
| | Feeling abnormal | 1 (2%) |
| | Pain | 2 (3%) |
| | Injection site discomfort | 1 (2%) |
| Injury, poisoning and procedural complications | Procedural pain | 15 (25%) |
| | Procedural dizziness | 7 (12%) |
| | Post procedural discomfort | 1 (2%) |
| Investigations | Acoustic stimulation tests abnormal | 1 (2%) |
| | Blood pressure increased | 1 (2%) |
| Musculoskeletal and connective tissue disorders | Pain in jaw | 6 (10%) |
| | Arthralgia | 1 (2%) |
| Nervous system disorders | Headache | 5 (8%) |
| | Dysgeusia | 4 (7%) |
| | Dizziness | 13 (22%) |
| | Dizziness postural | 1 (2%) |
| | Hypoaesthesia | 1 (2%) |
| Respiratory, thoracic and mediastinal disorders | Dry throat | 1 (2%) |
| | Oropharyngeal pain | 5 (8%) |
| | Throat irritation | 2 (3%) |
| Skin | Rash | 1 (2%) |
| | Acne | 1 (2%) |

Although our Phase IIa trial did not meet its primary endpoint of a ≥10 dB change in the average pure-tone air conduction threshold across 2, 4 and 8 kHz at 12 weeks, some patients showed positive changes in predefined secondary pure-tone audiometry and speech perception in noise measures. The Supplementary Information to this article provides additional data and a reflection on the use of DPOAEs in our trial. We feel that further investigation of the use of otoacoustic emissions and the choice of different stimuli such as new frequency primaries, short-pulse DPOAEs[22] and Stimulus-frequency OAEs, may prove valuable for inclusion in future trials for restoration of cochlear function.

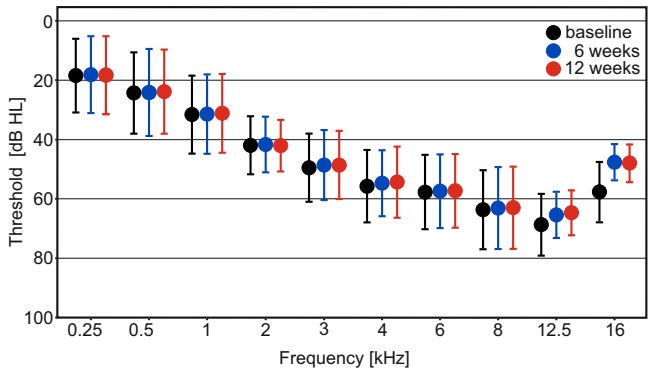

**Fig. 2 | Mean pure-tone air-conduction thresholds in the treated ear of all patients that completed 12 weeks follow-up (***N*** = 42).** Presented timepoints are baseline (black), 6 (blue) and 12 (red) weeks, error bars indicate standard deviations. Pure-tone thresholds are displayed in decibel hearing level (dB HL), frequency in kilohertz (kHz). Source data are provided as "supplementary data 4".

Our trial has generated important learnings about the design and delivery of early phase trials of novel hearing approaches. Through a detailed medical history and audiological assessment at the screening visit, checked against our in- and exclusion criteria, we aimed to include patients with SNHL most likely due to outer hair cell loss. We acknowledge that current tests of auditory function do not allow for deep phenotyping and therefore heterogeneity of our patient population in terms of underlying molecular mechanisms of SNHL may have diluted the effects of our highly targeted treatment. Collaborative efforts towards understanding the deep geno-phenotype of hearing and hearing loss are urgently needed[8,9,23].

We determined pre-trial stability of patients' hearing loss based upon their medical history and review of their previous audiograms. Eligibility was confirmed by the screening audiogram within 4 weeks of the first LY3056480 injection. One may argue that a pre-trial lead-in time with multiple audiological assessments would have ascertained stability of hearing loss of our participants.

We chose a traditional 3 + 3 dose escalation study design, starting with the 'no observed adverse effect level' dose from our pre-clinical studies and ending with the maximum dose based upon solubility of the formulation and middle ear volume, over an innovative Phase I design[24]. This is because at the time our trial was initiated, there were no data on the safety of novel hearing therapeutics. For future trials one may agree with regulators on more rapid dose escalation or integrated Phase I/II designs. For the same reason, we chose to inject one ear only. Now that more data are available on the safety of hearing regenerative approaches[20] and considering the observed outcomes in the untreated ear, we would recommend future trials adopt a placebo (diluent or saline) -controlled design, using patient-level randomisation.

At the time of development of the trial, we discussed various options for drug administration[25] among our Consortium and with our UK patient panel. At that time, patients shared a strong preference for intratympanic injections in an outpatient setting over a surgical approach that may require a general anaesthetic to deliver the drug directly onto the round window.

Due to the nature of our consortium and its public funding, our trial not only delivered on its drug development milestones, but also generated a wealth of long-term auditory measures for exploration of regeneration mechanisms. Such low-sample-size-high-dimensional datasets however have proven a challenge to existing statistical approaches and clinical interpretation. Going forward, machine learning may offer solutions to unlock these datasets with federated learning across trials.

We conducted the hearing tests as per local guidance, which explains why pure-tone audiometry was measured in 1 dB steps at the

German site and 5 dB steps at the UK and Greek site. Whether a smaller step size, improves accuracy and therefore detection of efficacy signals and may explain the differences across trial sites remains open for debate[26].

Finally, our trial team were contacted by more than 5,000 patients with hearing loss worldwide requesting to take part, illustrating the unmet clinical need this trial addresses.

## Methods

### Trial Oversight
The trial (ISRCTN59733689, registration date 16/5/2017) was designed and coordinated by the REGAIN Consortium (https://www.regainyourhearing.eu), supported by an EU Horizon 2020 grant (https://cordis.europa.eu/project/id/634893), sponsored by the Consortium lead Audion Therapeutics BV and overseen by an independent steering committee. The trial protocol was approved by national regulatory authorities, medical ethical committees and local R&D authorities in London (London, Central REC Committee, REC Number: 17/LO/0632), Athens (Hellenic Republic Ministry of Heath National Ethics Committee - Reg. No.: 83052/2018) and Tübingen (Ethik-Kommission der Medizinischen Fakultät und am Universitätsklinikum Tübingen, Proj. No.: 592/2018AMG1). The trial was conducted in accordance with GCP and principles of the Declaration of Helsinki and enrolment first started after trial registration and ethical approval Patient and study data were monitored by an independent Contract Research Organisation. During the Phase I trial, an independent Data Safety Monitoring Board (DSMB) assessed safety and tolerability (see Supplementary Information page 4).

### Patients
Eligible for the Phase I/II trial were patients aged from 18 to 80 years with a primary complaint of hearing loss of more than 10 years duration and stable hearing, their history suggesting an age-related, noise-induced or unknown cause, and diagnosed at the trial screening visit with bilateral symmetrical (<15 dB difference between ears) mild to moderate SNHL (mean pure-tone hearing threshold 25 to 60 dB HL across frequencies 0.5, 1, 2, 4 and 8 kHz). Patient sex was self-reported. Excluded were patients presenting with a primary complaint of tinnitus, a 'true' air-bone gap >15 dBHL in 3 or more contiguous frequencies between 0.5, 1, 2, 4 kHz, a history of suspected or diagnosed genetic cause of hearing loss, suspected or known diagnosis of inner ear pathology (see Table S1 in the Supplementary Information for details), evidence of acute or chronic middle ear disease and/or surgery), use of ototoxic medication within 12 months of screening, or ongoing or planned systemic or local drug-based therapy for SNHL or tinnitus during the study.

### Trial Procedures
This Phase I/ IIa trial was open-label so there was no randomisation or blinding. Patients either self-referred to the local trial teams via the REGAIN website https://www.regainyourhearing.eu/ or were referred by ENT surgeons or audiologists from hospitals serving as Participant Identification Centres (PICs). Trial teams pre-screened potential participants over the phone and gave information about the trial.

Those potentially meeting the inclusion criteria were invited to a screening visit at the trial hospital sites where informed written consent was taken for the trial, and hearing and balance assessments as well as vitals sign measurements were conducted to verify eligibility (see Table 3). Patients were not compensated for participation in the study, except for reimbursement of travel expenses.

For the Phase I trial in UK, a Cone Beam CT scan of the temporal bone was made to explore accessibility and permeability of the round window to the study drug. Results were classified into 3 groups: no, up to 50%, and 50-100% opacification of the RWN, where the last group would be excluded from the trial. Because no patients met this

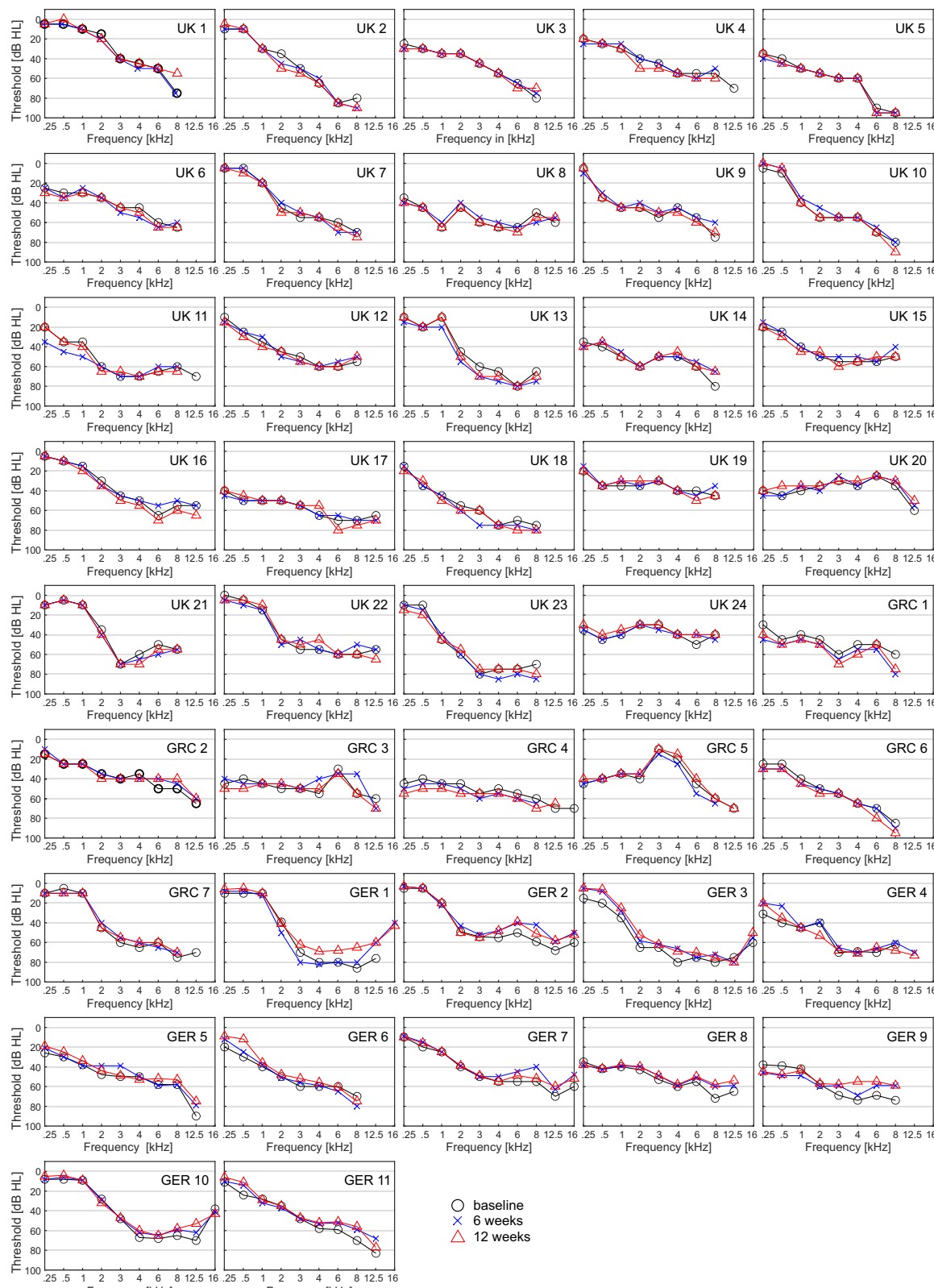

**Fig. 3 | Individual pure-tone audiograms (treated ear) of all patients that completed 12 weeks follow-up (*N* = 42).** Presented timepoints are baseline (black circle), 6 (blue cross) and 12 (red triangles) weeks. Frequencies not shown indicate that thresholds are beyond the maximum output of the audiometer. Pure-tone thresholds are displayed in decibel hearing level (dB HL), frequency in kilohertz (kHz). Source data are provided as "supplementary data 4".

**Table 3 | Table of Assessments**

| Activity Visit window | Screening | Treatment period | | | | Follow up period | | Optional visits | |
|---|---|---|---|---|---|---|---|---|---|
| | | Day 1 | | Day 8 | Day 15 | Week 6 | Week 12 | Month 6 | Month 12 |
| | | Baseline | Dose 1 | Dose 2 | Dose 3 | | | | |
| Informed Consent | X | | | | | | | | |
| Review of in/exclusion criteria | X | X | | | | | | | |
| Medical history and demography | X | | | | | | | | |
| Vital signs | X | X | X | X | X | X | X | | |
| Physical examination | X | X | | X | X | X | X | X | X |
| 12-lead ECG | X | | X | X | X | X | X | | |
| Laboratory assessments | X | | X | X | X | X | X | | |
| Clinical balance assessment | | | X | X | X | X | | X | X |
| Full balance assessment | X | | | | | | X | | |
| Facial nerve function | X | | X | X | X | X | X | X | X |
| Taste assessment | | | X | X | X | X | X | X | X |
| Otomicroscopy | X | X | | X | X | X | X | X | X |
| PTA | X | | X | X | X | X | X | X | X |
| Speech Audiometry | X | | | | | X | X | X | X |
| DPOAE, SNR and absolute | | X | | | | X | X | X | X |
| Cochlear dead region assessment | | X | | | | X | X | | |
| ART | | X | | | | | X | | |
| Tympanometry | X | | | | | | X | X | X |
| Tinnitus assessment | X | X | X | X | X | X | X | X | X |
| QoL (HHIA/E questionnaire) | X | | | | | X | X | X | X |
| Hearing Aid questionnaire | X | | | | | | | X | X |
| Drug dosing - Injection | | | X | X | X | | | | |
| Pregnancy testing | X | X | | X | X | X | X | | |
| Concomitant medication | X | X | X | X | X | X | X | X | X |
| Adverse Events | X | X | X | X | X | X | X | X | X |

exclusion criteria in the Phase I trial, this investigation was taken out of the Phase IIa trial protocol.

At the baseline visit, after confirming eligibility, patients were treated with the study drug in one ear, i.e., the poorer hearing ear according to pure-tone and/or speech audiometry, in case of no difference patients were asked to identify their poorer hearing ear, and if neither applied the ear best accessible for injection was chosen. Under otomicroscopic visual control lidocaine/prilocaine cream (EMLA® cream) was applied onto the tympanic membrane to achieve local anaesthesia; the cream was removed after 30-45 minutes using microsuction. With the patient in supine position 0.5 ml of the study drug was injected into the anterior inferior quadrant of the tympanic membrane with a 25/26-gauge needle and bevel facing in an inferior posterior direction to a depth of 2-3 mm just inferior to the round window niche. Patients remained supine for 30 minutes, their head turned 45 degrees towards the treated ear and advised to not talk, sneeze, and cough. They were monitored for safety for 24 hours after the first injection of the study drug in the Phase I trial and for 4 hours after the second and third injection in the Phase I and all injections in the Phase IIa trial.

In the single-site (UK) Phase I first-in-human trial (from January 24 to October 17, 2018), consecutive cohorts of 3 patients received 3 intratympanic injections of LY3056480 at a specified dose level (ascending doses of 25, 125, 200, 250 µg across cohorts), one week apart in one ear. After dosing each cohort, the Data Safety Monitoring Board assessed the safety and tolerability of the study drug and procedure. In addition, the protocol stated that the trial (Phase I and IIa below) may be discontinued at the discretion of the Coordinating Investigator, Principal Investigator, Sponsor or Independent Ethics Committee based on the occurrence of the following (but not limited

to): AEs unknown to date with respect to their nature, severity, and duration; increased frequency and/or severity and/or duration of AEs; medical or ethical reasons affecting the continued performance of the study; cancellation of drug development; notification by regulatory authorities.

In the Phase IIa trial across 3 tertiary care otology services (UK, Greece, Germany; from January 30 to August 5, 2019) all patients received 3 intratympanic injections of 250 µg LY3056480 one week apart in one ear.

Patients were followed up after 6 and 12 weeks, with an optional long term follow up visit at 6 and 12 months. During all trial visits a broad repertoire of hearing assessments were performed: pure-tone audiometry, speech-in-noise, middle ear immittance, DPOAE and TEN test, balance tests (eye movements, head thrust, modified Romberg, Unterberger, bithermal air calorics), quality of life measurements (HHIA/E, TFI, DHI) and general safety assessments (facial nerve and taste function, laboratory, vital signs, ECG). The full trial protocol is provided as "Supplementary Data 1" (study protocol version 1.0) and "Supplementary Data 2" (study protocol version 5.0).

**Outcomes**

As per the trial protocol (see Supplementary Information) the primary outcome of Phase I was safety and tolerability in terms of occurrence and severity of treatment and procedure-related local and systemic Adverse Events (AEs). The secondary outcome was the optimal dose of LY3056480 for Phase IIa. Local safety outcomes included changes in hearing in the treated ear, tinnitus, balance and facial nerve function. Systemic safety outcomes included vital signs, haematology, chemistry and electrocardiography. Adverse Events were graded according

to Common Terminology Criteria for Adverse Events (CTCAE) version 5.0[27]. The same safety outcomes were assessed in Phase IIa.

The Phase IIa primary efficacy outcome was change in the average pure-tone air-conduction threshold (in dB HL) across 2, 4 and 8 kHz in the treated ear from baseline to 12 weeks. Secondary efficacy outcomes were changes from baseline to 6 and 12 weeks for both the treated and untreated ear in: (1) the average pure-tone air-conduction threshold (in dB HL) across 2, 4 and 8 kHz; (2) pure-tone air-conduction thresholds at individual frequencies (dB HL), including 12.5 and 16 kHz; (3) speech reception threshold in noise, expressed as the SRT50n which is defined by the signal-to-noise ratio (SNR) that yields an average response of 50% correctly recognised words; (4) distortion product otoacoustic emissions (DPOAE) mean amplitude and SNR, for both low-level (65/55 dB SPL) and high-level (70/70 dB SPL) tone primaries; (5) Hearing Handicap Inventory for Adults/Elderly (HHIA/E) score (per participant).

## Audiological assessments

Pure-tone audiometry was conducted at 0.25 kHz, 0.5 kHz, 1 kHz, 2 kHz, 3 kHz, 4 kHz, 6 kHz, 8 kHz, 12.5 kHz and 16 kHz, following a 'down-10/up-5' dB rule (5 dB step-size) in the UK (Otometrics Madsen Astera 2; Sennheiser HDA 300 circum-aural high-frequency headphones; Bone-Conductor Otometrics BC-71) and Greece (Interacoustics Affinity; circumaural Headphones DD45; Sennheiser HDA 300 (circum-aural high-frequency headphones); Bone-conductor Radioear B71) and a down-10/up-1 rule (1 dB step-size) in Germany (AT1000 Auritec; Sennheiser HD300 circum-aural high-frequency headphones; Bone-Conductor Radioear B71W), as per local guidelines. Threshold was defined as the lowest decibel hearing level at which responses occur in at least one half of a series of ascending trials. The minimum number of responses needed to determine the threshold of hearing was two responses out of three presentations at a single level. Blinding of the Audiologists to previous audiometric results was not included in the study design.

Speech perception in noise was measured with a words-in-babble test (WiB) at all sites[28]. Care was taken that the same word lists were not presented at each visit. In UK, the words-in-babble test used lists of 25 monosyllabic meaningful English words as targets, the masking noise was a multi talker babble (Otometrics Madsen Astera 2). The test was presented monaurally in a sound-proof room on a calibrated computer using custom-written Matlab software with Sennheiser HD 125 headphones. The SNR was varied adaptively during the test, with the level of the masker fixed at about 65 dB SPL and the level of the target speech varied. The initial SNR was +20 dB SNR and was decreased after each single correct response (i.e., increasing difficulty) or increased after each incorrect response (i.e., decreasing difficulty) ± 2 dB SNR. The test stopped either at 8 reversals or 25 words. A threshold value was then calculated as the mean of the final six to eight reversals, which is the SNR needed for a performance level of about 50% correct, known also the Speech Reception Threshold (SRT50n).

In Germany, the WiB test used lists of 20 monosyllabic meaningful German words as targets (Freiburger Test), the masking noise was a multi-talker babble. The test was presented monaurally in a sound-proof room on an Auritec AT1000 (Auritec) with Sennheiser HD 125 headphones. The level of the masker was fixed at 65 dB SPL and the level of the target speech varied, starting with the initial SNR of 20 dB. To determine the SRT50n, at least 2 lists of each 20 words were tested at different levels, and SRT was calculated from a regression of the two test lists that were just below and just above the 50% intelligibility.

In Greece, the WiB test used 4 lists of 50 disyllabic phonemically balanced words and multi-talker babble as the masking noise. The test was presented monaurally (Interacoustics Affinity with DD45 headphones) in a sound-proof room. Each of the 4 lists were presented in a random order. The signal intensity level remained constant at 20 dB over the SRT while the babble noise intensity was varied. Each ear was

presented with a total of 9 noise levels (−6 dB, −3 dB, −1 dB, 0 dB, +1 dB, +3 dB, +6 dB, +9 dB, + 12 dB), always with this order and starting from the most demanding presentation (−6 dB) to reduce the learning effect. The SRT50n was derived directly from the performance intensity (PI) function.

DPOAE measurements were carried out for both low-level (65/55 dB SPL) and high-level (70/70 dB SPL) tone primaries on the Madsen Capella2 (Natus Medical) in UK and Germany and on TITAN (Interacoustics) in Greece. Settings were: F2/F1 Ratio 1.22; 8 Bands per octave, 3 Blocks, 90 sweeps and 5 retries. At the beginning and at the end of each measurement, the correct fit of the probe in the ear was verified by the device software according to the manufacturer's recommendation.

## Statistical analysis

The number of participants in Phase I was determined by its 3 + 3 design. For Phase IIa we set a recruitment target of 40 patients, based on 87% power to detect a 10 dB change (standard deviation 20 dB) corresponding to an effect size of 0.50[29].

Data was collected through an eCRF, for Phase I via Open Clinical and for Phase IIa via Castor. Analyses were performed with SAS (v9.4), SPSS (v26), and R (v3.3.1).

As per protocol, all data collected and available was used in the analysis. No imputation of missing data was applied. See the Statistical Analysis Plan (SAP, see "Supplementary Data 3") for details on handling of missing data. The detailed SAP was finalised before database lock.

All Phase IIa analyses were performed according to modified intention-to-treat, including patients with pure-tone audiometry data at baseline and at least once post-LY3056480 administration. The statistical significance level was set at 0.05. Bonferoni correction was used to control for family-wise error rate.

Phase IIa efficacy outcomes were analysed separately for treated and untreated ears (except for Hearing Handicap Inventory for Adults/Elderly) and for the total number of participants across the three trial sites as well as per trial site. A linear mixed-effect model was used to account for repeated measures and the multilevel structure of the pure-tone audiometry, speech-in-noise data and DPOAEs. Patient age, baseline audiometric values, follow-up timepoint, and timepoint-by-treatment interaction were entered as fixed factors. Patient (random intercept) was entered as random factors. An unstructured covariance matrix was used.

See Supplementary Information for the post-hoc exploratory analyses performed for pure-tone audiometry and speech perception in noise measures at the optional follow-up visits 6 and 12 months and for results per trial site.

For pure-tone audiometry we analysed the mean change from baseline to 6 and 12 weeks in the average pure-tone threshold across 2, 4, and 8 kHz, as well as at individual frequencies, including 12.5 and 16 kHz.So far, very few studies of hearing regenerative drugs have been conducted and definitions of clinically important hearing improvement in terms of pure-tone thresholds have not been agreed upon. Some guidance can be deducted from the 1994 American Speech-Language-Hearing Association (ASHA) Guidelines for the Audiologic Management of Individuals Receiving Cochleotoxic Drug Therapy[30], where a hearing change (in this case decrease) is defined as a 20 dB decrease at any one test frequency; or a 10 dB decrease at any two adjacent test frequencies. Likewise, Campbell et al. [31] define a significant and clinically relevant noise-induced threshold shift as ≥10 dB at one or multiple test frequencies. We therefore also analysed at 6 and 12 weeks the number and percentage of participants showing: (a) a positive change of ≥10 dB in pure-tone air conduction- threshold at any frequency, and (b) a positive change of ≥10 dB in pure-tone air-conduction threshold in two or more adjacent frequencies, or ≥20 dB in a single frequency.

For speech perception in noise we analysed the mean change in SRT50n (see above) from baseline to 6 and 12 weeks. For DPOAEs we

analysed the mean amplitude and SNR, for both low-level (65/55 dB SPL) and high-level (70/70 dB SPL) tone primaries as well as the change in overall distribution of DPOAE categories 'present-normal, present-abnormal, and absent' from baseline to 6 and 12 weeks, using the MacNemar test.

HHIE and HHIA scores were combined rather than analysed separately, as the scoring mechanisms are the same and have been tested for internal consistency reliability. We analysed the mean change in score from baseline to 6 and 12 weeks, applying the McNemar test.

Differences between the treated and untreated ear in change from baseline to 6 and 12 weeks were expressed as numbers of patients with a difference of >10 dB in one or more frequencies for pure-tone hearing threshold across 2, 4, and 8 kHz. No statistical analysis was performed on these numbers.

### Reporting summary

Further information on research design is available in the Nature Portfolio Reporting Summary linked to this article.

### Data availability

The latest version (5.0) of the study protocol, the statistical analysis plan and source data file with demographics, adverse events and pure-tone audiometry data are provided in with this manuscript and its Supplementary Information. Additional Data supporting the current trial findings can be made available from the REGAIN Consortium (Rolf Jan Rutten, rjrutten@audiontherapeutics.com, Audion Therapeutics BV, Amsterdam, the Netherlands). All study data will be shared from the publications onwards until 1 year after publication in a password protected controlled environment, including individual deidentified participant data. Responses will be sent within 2 weeks. Restrictions apply to the availability of these data due to planned regulatory submission. Source data are provided with this paper.

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

### Acknowledgements

Horizon 2020, the EU Research and Innovation programme, funded the REGAIN project (ISRCTN number 59733689). Audion Therapeutics BV funding supported long-term follow up of trial participants. We thank

Kim Airey, George Shaya, Marta Merida, Glyn Ang, Rishi Mandavia, Nishchay Mehta, Joseph Manjaly, Robert Nash, Andrew Hall, Joanne Palmer, Tanjjnah Ferdous, Natallia Kharytaniuk, Royal National ENT Hospital nursing team, Lamprini Agrapida, Katharina Thum, and Andreas Heyd for their contribution to the REGAIN trial. We are grateful to the participants of this trial whose motivation and support were invaluable to the success of the REGAIN project. We are grateful to the participants of this trial whose motivation and support were invaluable to the success of the REGAIN project.

## Author contributions

A.G.M.S. is the coordinating investigator, S.S., S.W. and A.G.B. the principal investigators at the 3 sites. A.G.M.S., S.W., S.S., A.S.B.E., H.B., L.M.M. and A.G.B. designed and planned the trial and its data analysis and interpreted the results. K.P. and A.G.B. performed statistical analyses. The first draft of the manuscript was prepared by A.G.M.S., drafts were reviewed and edited by all authors. All authors made the decision to submit the manuscript for publication and vouch for the accuracy and completeness of the data and for the fidelity of the trial to the protocol.

## Competing interests

A.G.M. Schilder advises hearing industry on clinical trial design and delivery. A.S.B. Edge is a consultant and shareholder of Audion Therapeutics BV. L.M. Middelink is consultant to Audion Therapeutics BV. The remaining authors declare no competing interests

## Additional information

[1]National Institute for Health Research University College London Hospitals Biomedical Research Centre, London, UK. [2]Ear Institute, University College London, London, UK. [3]Royal National ENT and Eastman Dental Hospitals, University College London Hospitals Trust, London, UK. [4]Department of Otolaryngology, Head and Neck Surgery, University of Tübingen, Tübingen, Germany. [5]Middelinc, Utrecht, the Netherlands. [6]Department of Otolaryngology, Harvard Medical School, Boston, USA. [7]1st Department of Otolaryngology, Hippocration Hospital Athens, National & Kapodistrian University of Athens, Athens, Greece. ✉e-mail: stephan.wolpert@med.uni-tuebingen.de

## REGAIN Consortium

**Anne Schilder[1,2,3], Stephan Wolpert [4]✉, Shakeel Saeed[1,2,3], Leonie Middelink[5], Albert Edge[6], Helen Blackshaw[1,2], Kostas Pastiadis[7], Athanasios Bibas[7], Elizabeth Arram[1,2,3], Asger Bilhet[8], Hannah Cooper[1,2], Ernst Dalhoff[4], Femke van Diggelen[9], Rolf Jan Rutten[10], Helmuth van Es[10], Karin Hojgaard[1,2,3], Eleftheria Iliadou[7], Omursen Yildirim[1,2,3], Sherif Khalil[3], Dimitris Kikidis[7], Hubert Lowenheim[4], Nikos Markatos[7], Marcus Mueller[4], Thore Schade-Mann[4], Fritz Schneider[4], Katerina Vardonikolaki[4] & August Wilke[11]**

[8]Nordic Bioscience, Herlev, Denmark. [9]TTopstart BV, Utrecht, the Netherlands. [10]Audion Therapeutics BV, Amsterdam, the Netherlands. [11]Eli Lilly and Company, Indianapolis, USA.

