## [Peer Review File · Nature Communications]

REVIEWER COMMENTS

Reviewer #1 (Remarks to the Author):

As the authors outline, hearing loss is among the most prevalent untreated disabilities worldwide. Currently, there are no approved pharmacological therapies that successfully prevent/reverse cochlear sensory hair cell loss. Thus, the area of study is very worthwhile. This study demonstrates safety and treatment tolerance of the investigational drug when administered via transtympanic injection. However, none of the data suggest clinically significant therapeutic benefit. When considering test-retest pure tone variability (10 dB), the outcome for individual frequencies should have been >10 dB. If the authors believe the small changes seen in select patients, a randomized arm incorporating saline injection (no drug) would be beneficial. The conclusions are overstated. The data presented in the study do not indicate therapeutic benefit.

Reviewer #2 (Remarks to the Author):

Schilder et al., present the results of phase I and IIa of the REGAIN clinical, a trial to restore hearing in individuals with long-standing, stable, sensorineural hearing loss (SNHL). Specifically, they use gamma secretase inhibitors (GSI) that inhibit the Notch pathway. The study was conducted as a multi-institutional clinical trial, using three different sites, and three different languages for the audiometric analyses. Results are presented for safety and toxicity as well as for efficacy. Study results are reported, and the authors conclude that 'GSI is safe and can lead to clinically relevant improvements in multiple measures of hearing function'. Careful review of the manuscript reveals significant concerns for bias in reporting study rationale and results. The reviewers do not agree with the presentation and summary of the data, discussion and conclusions. Below is a point-by-point summary of concerns.

Major comments:

1. The authors base the study on the potential of 'GSI to induce mammalian OHC regeneration and partial hearing restoration' which is also the first sentence of the abstract. Review of the manuscripts that were used as the rationale for the study reveals that there is no evidence to support efficacy of the proposed approach in adults with a stable hearing loss. The evidence in the literature indicates that GSI result in hair cell regeneration in P1 neonatal/cultured cochleae or in acutely deafened animals. The only three papers that are cited to support the use of this approach in adults include: (a) Hori et al – showed that only in acutely damaged guinea pig ears from ototoxic drugs, very few OHC can regenerate (and only in the non-sensory domain); (b) Mizutani et al – show that treatment of mice with GSI acutely after noise exposure can result in some improvement in the permanent threshold shifts in the low but not high frequencies. The same group reported in meetings that delayed treatment does not work; (c) Tona

et al – show that guinea pigs acutely treated with GSI (within 7 days of noise exposure) had partial hearing restoration. Thus, there is no evidence in the literature to show that without acute injury in a mature inner ear – there would be any response to GSI treatment. Further, Hori et al show no response when the ototoxic treatment was incomplete. Finally, a paper from 2016 by the laboratory of Dr. Groves (27918591) shows that GSI inhibition results in no molecular response or changes in cochlea that are older than P6 in mice ('Mouse cochlear supporting cells become almost completely unresponsive to blockade of Notch signaling by six days of age'). One could argue that the authors wanted to try this approach despite the lack of proper support for it in the peer reviewed literature. In this case, the peer reviewed literature must be presented appropriately and the failure to reach statistically significant endpoints needs to be explained also in the context of the conflict between the study design (adults with stable hearing loss and no acoustic/ototoxic trauma) and hints for possible efficacy in the literature (neonatal ears or acutely injured ears in adults). In particular, the first sentence of the abstract should be changed to end with 'in the setting of acute injury only', the introduction should be modified to reflect these conflicts, and the discussion should be modified to address a rationale for a possible negative result of the study.

2. Partial reporting of adverse effects. The safety outcomes are reported only in a table without providing any detail on the most common adverse effects. Specifically, tinnitus is reported as experienced by 46% of the patients. However, no information is reported as for the duration of the tinnitus, whether the patients that reported tinnitus had tinnitus also before the study, whether the tinnitus was in the treated or control ear (specifically as the authors claim some improvement in hearing in the contralateral ear), and long term duration of tinnitus (according to IRB table 4, TFI is collected for all patients in weeks 6 and 12 of the study).

3. Reporting of improvement in PTA. The reporting of improvement in PTA revealed no change in the 2,4, and 8 kHz thresholds. It is not clear why the classic 500, 1000 and 2000 PTA are not also reported, specifically when animal studies suggested that efficacy, if present, is primarily in the lower frequencies. Furthermore – the PTA at these classic frequencies is the more important PTA for speech understanding. The authors proceed to report changes in hearing thresholds per individual frequencies, and find some improvement in 45%, 50% and 44% in one frequency at 3, 6 and 12 months, respectively. It is not clear if the improvement is in the same patient across tests. The data for the contralateral ear is not summarized in the context of individual patients, and fig 2 patient C shows improvement in the contralateral ear. The data should also be presented divided to frequencies 2kHz and lower (most relevant speech frequencies), and 4kHz and above (less relevant to speech understanding and most susceptible to test-retest variability from calibration or microphone placement), longitudinally per ear, and in comparison, to the other ear.

4. In the section on outcomes for the untreated ear: pure tone improvements of 10 dB or greater for one or more frequencies are reported for 44-62% at the various time points. How many of these persons had improvements at a single frequency? How many had stable hearing at all frequencies? How many had

hearing decline of 10 dB at one or more frequencies? How many had an improvement at some frequencies and a decline at other

5. There are several concerns about missing data that give the impression of selective reporting of positive findings. For example, the authors report trial site specific outcomes and limit the description of the pure tone findings to the German site whereas the comparable findings for the UK and Greek cohorts are not mentioned. Does this mean there was not a statistically significant improvement in pure tone thresholds for the subjects tested at the UK and Greek sites? Why is that not presented? Do the authors have an explanation for site differences?

6. The authors do not provide enough detail about interpretation of the OAE data. Validity of the OAE must be taken into account using criteria such as those presented by Reavis et al. 2011 (PMID: 20625302) that consider a minimum level of noise, a minimum OAE amplitude, and a minimum SNR. When these criteria are not met, the OAE is considered absent. Given the degree of hearing loss required for study eligibility, it is expected that OAEs were absent at some or many frequencies. How was the absent OAE handled? Was this treated as missing data? Were the collected levels used without consideration of response validity? Was an arbitrary number assigned to the absent responses?

7. OAE data shown in the supplemental tables show a number of frequencies where the SNR improved in the absence of a concurrent change in OAE amplitude itself at that same frequency. This implies that the noise levels were lower and suggests that these SNR changes do not represent improvement in cochlear function, but instead a reduction in environmental or physiologic noise during data collection.

8. Do the authors have an explanation for the statistically significant effect of study site ("clinic") on the OAE changes as shown in the supplemental tables.

9. I struggle with the statistically based definition of a clinically relevant change in OAEs and recommend that the authors examine the literature on OAE stability and derive an evidence-based definition of a significant change in the OAE. This article, that provides guidance on serial monitoring of OAEs in clinical trials should be considered: Konrad-Martin D, et al, 2016, PMID: 27518137.

10. Discussion - the study, overall, had a negative result, and the discussion should be modified to reflect that and focus on it with a minor discussion of possible positive outcomes if the study would be heavily redesigned, rather than a focus on positive outcomes that are not founded in the presented data.

Methods comments:

1. In a study that has the potential to inform therapeutic interventions for hearing loss, it is important that the methods section contain enough information to allow independent replication. The current description in the manuscript, the IRB document, and the other supplementary materials falls short. For example, were the audiologists blinded to previous hearing test results? If not, this is a potential bias. Were the earphones insert, supra-aural or circumaural? Did this vary between tests? What was the presentation level for the words in noise testing? Was care taken to ensure that the same word lists were not presented at each visit. What parameters were used to collect DPOAE data (number of sweeps, F1/F2 ratio?) What procedures were used to ensure good fit of the OAE probe and acceptable signal in the ear canal?

2. The initial audiogram, which was obtained 0-14 days prior to the intervention, was used to determine eligibility, and serve as baseline. Stable hearing prior to treatment is important consideration for subject participation in a study aiming to show an effect of intervention, and this is one of the exclusionary criteria for subject enrollment in this trial (IRB supplement). However, there is no mention of methods used to ensure hearing stability and no replication of the pre-treatment audiogram. This is a shortcoming.

3. There are a number of criteria used to define a significant change in pure tone thresholds. This includes the widely used and accepted ASHA (1994) definition that requires a 10 dB change at two consecutive frequencies, or a 20 dB change at a single frequency. The authors of the current manuscript define a significant change in hearing based on the pure tone average of 2, 4 and 8 kHz as an improvement ≥ 10 dB and, mathematically, this would achieve the ASHA criteria. However, the authors subsequently report improvement in hearing by 10 dB at one or more frequencies. An improvement in hearing at a single frequency by 10 dB is not acceptable evidence and it is recommended that the authors re-evaluate their data using criteria such as those established by ASHA.

4. What is the basis for the 2dB change in the speech in noise SNR as the cut point? Provide references and a rationale based on the known psychometric functions of each test used to evaluate speech in noise listening.

Minor comments –

1. The authors address hearing loss as a single disease and group patients with different durations of hearing loss, different severities of hearing loss, and different audiogram configurations as one group. This type of approach assumed a 'one size fits all' for a treatment that strongly decreases that chances of identifying successful outcomes given the heterogenous nature of hearing loss.

2. Discussion – the authors explain that hearing was measured up to 16 kHz as higher frequencies are important for understanding speech in noise. 4 and 6 kHz are important for speech in noise, but the reviewers are not familiar with literature showing that 12-16 kHz (which are more variable) – have any effect on speech in noise. Furthermore, for none of the patients had the hearing handicap improved.

3. Discussion – ‘going forward we would recommend future trials to assess pure-tone hearing thresholds as 1-2dB(HL) steps... matrix test’. Please remove this sentence. There are no data to support this claim in the results. Furthermore, there are no data to show clinical significance of 1dB. Assessing hearing at steps that are clinically insignificant and are at a significant risk of test-retest variability will muddy the literature and result in irresponsible use of resources.

4. ‘our findings support our hypothesis and suggest DPOAEs can be used to detect HC regeneration’ – please remove this sentence. There results do not support this claim.

5. ‘next steps could include testing of efficacy in higher dosages’ – the presented data do not support this suggestion. Perhaps restructuring the study to address instances of acute hearing loss would be more appropriate and grouping patients by type and duration of hearing loss. However, the current study does not support efficacy.

6. Discussion – closing sentence – please modify as conclusion is not supported by data.

7. Discussion – improvements in the contralateral ear: this discussion should be completely removed. There is no quantification or rigorous study of hearing changes in the contralateral ear. The proposed mechanisms by the group are highly unlikely. And if this remains, error and issues with test retest reliability at the frequencies that the authors do detect improvement should be discussed and considered as well .

Reviewer #3 (Remarks to the Author):

This is a proof-of-concept in a Phase I/IIa multiple-ascending-dose open-label trial to establish the safety and efficacy of a regenerative drug treatment for sensorineural hearing loss. Well written paper and well design study. The primary endpoints were pure-tone audiometry at 3 months. There is no serious concern on the safety and tolerability, but improvement was not observed and maximum tolerated dose was not met. I have only comments on the clarity of the design:

- CONSORT: in phase 1, could you please check the consistency as 12 patients were not enrolled but justification were given for 13 patients
- A sample of 15 is quite small for a 3+3 design, could the authors clarify the design and the choice of 250 microgram, what was the first dose 25 or 125 or 200 or 250 microgram? A diagram of safety results by cohorts will be more informative than a table.
- Safety and tolerability endpoints in the SAP (6.2.3 paragraph) are 6 endpoints, these need to be all reported in a supplement table as planned.
- The primary outcome was well investigated, but the secondaries need to be reported as stated in the SAP for consistencies.
- Discussion: please elaborate on the dosages and tolerability of higher dose and the progression to phase IIa, given 250 microgram seems safe but maybe not MTD. For example would you re-do the phase 1 study with higher doses?

Reviewer #4 (Remarks to the Author):

The paper by Schilder et al. written on behalf of the REGAIN consortium reports on safety and efficacy of LY3056480 in patients with sensorineural hearing loss (SNHL) in phases I and IIa studies. The authors claim to have “delivered proof-of-concept in a Phase I/IIa multiple-ascending-dose open-label trial”. However, the authors also confess that “the primary endpoint of >10 dB (HL) improvement across 2, 4, 8 kHz was not met”. This means that the REGAIN study did not deliver proof-of-concept as defined in the study protocol and ought to be considered negative. The careful language describing the findings is critical because humans suffering from hearing loss are expecting positive news from clinical research on means expected to help them.

Other critical issues which were not addressed adequately by the authors are:

- 1) German center delivered ample improvement of audiologic biomarkers whereas both the British and Greek centers did not at all. There is no explanation offered for such disparity.

2) Safety and tolerability of the LY3056480 may be confirmed by the phase I study. However, there is little attention placed on administration procedures. That is critical for multiple drug administration and compliance. It has not been critically discussed and no alternatives have been suggested.

3) In addition, a clear-cut definition of adverse events that suggests toxicity from the investigated drug (DLT, dose-limiting toxicity) is missing. It is therefore unclear, if the targeted probability of the 3+3 for a DLT is appropriate. It is unfortunate that doses higher than 250 µg were not considered in the phase I trial.

4) The authors state that higher efficacy could be achieved by elevating the dose of LY3056480, however the maximum dose tested in the phase I was 250 µg. This dose cannot be increased due to regulations unless another phase I study is performed. In other words, increasing the dose in order to increase efficacy based on the results of the current phase I study should not be suggested. The only way to increase the dose is by increasing the number of injections at a maximum tested dose of 250 µg/injection over time.

This study will profit from applying objective interpretation of the data as regulatory agencies require. The report, as stands, is delivering a message of success. However, the data do not support that message, particularly not in case of efficacy.

The REGAIN trial is leading to the follow up trial named VESTA. VESTA is a double blinded, randomized, placebo controlled, multi-center efficacy phase II trial comparing three different dosing regimens of LY3056480 based on the maximum single dose of 250 µg explored in the REGAIN trial. Sufferers from mild to moderate SNHL will be recruited. Four injections of 250 µg of LY3056480 or placebo will be administered trans-tympanically into one ear. The primary outcome measure of efficacy of local treatment with LY3056480 in terms of hearing after 6 months will be the number of responders to therapy with at least a 2 dB improvement in an adaptive manner in noise test compared to placebo. All this is a result of experience gained from the REGAIN trial but these conclusions are not explained in the current report.

There is no criticism regarding quality of study protocol, procedures employed, or statistical evaluation of the data. However, the 3+3 design might be a little old-fashioned, but might be the easiest to implement under complicated study conditions. In addition, since the study does not have control group, it is not randomized. This may be seen as limitation, since in pre/post comparisons, the observed effect may be considered as higher.

Remarks from Reviewer #1

As the authors outline, hearing loss is among the most prevalent untreated disabilities worldwide. Currently, there are no approved pharmacological therapies that successfully prevent/reverse cochlear sensory hair cell loss. Thus, the area of study is very worthwhile. This study demonstrates safety and treatment tolerance of the investigational drug when administered via transtympanic injection.

- a. However, none of the data suggest clinically significant therapeutic benefit. When considering test re-test pure tone variability (10 dB), the outcome for individual frequencies should have been >10 dB.*
- b. If the authors believe the small changes seen in select patients, a randomized arm incorporating saline injection (no drug) would be beneficial.*
- c. The conclusions are overstated. The data presented in the study do not indicate therapeutic benefit.*

Author response

- a. See also Reviewer #2 methods comment 3). We value this feedback, have performed additional analyses to address this comment based upon a thorough review of the literature and have amended the manuscript Methods section (page 19, lines 438- 451). We therefore defined individual responses pragmatically as: 1) a positive change of ≥ 10 dB in pure-tone air conduction threshold at any frequency, and 2) a positive change of ≥ 10 dB in pure-tone air conduction threshold in two or more adjacent frequencies, or ≥ 20 dB in a single frequency. In addition, we defined consistent hearing improvement over time as a positive change of ≥ 10 dB at a certain frequency which was also present at two or more consecutive follow-up visit. The results of these analyses have been added to the manuscript Results section (page 7, lines 143-148).
- b. The Reviewer raises an important point. When we engaged with the Medicines and Healthcare products Regulatory Agency (MHRA) in 2016 to start the approvals process, we were the first team worldwide to trial a regenerative hearing drug applied through an intratympanic injection. Since we could not rely on existing evidence regarding the safety of such an intervention, we agreed with MHRA to approach this like we would in clinical practice for patients with bilateral ear conditions requiring surgery and in the presence of residual hearing. Due to the risk of surgery to that residual hearing, ENT surgeons operate on one ear at a time, starting with the poorer hearing ear and deciding on surgery on the better hearing ear only after recovery from surgery in the poorer hearing ear. In our trial we therefore decided to inject one ear only (the poorer hearing ear) and avoid any procedure to the other ear. Now there is more information from our trial and other trials of locally applied regenerative hearing drugs, we agree that a placebo-controlled trial design, either with the diluent only (no active compound) or saline, would be acceptable to regulators and ethics committees. From our observations in the untreated ear, we would recommend these trials would use patient-level randomization.
We have amended our discussion (pages 11, lines 229-232).
- c. We agree that our conclusion may have been read as an overstatement. We have therefore amended the conclusion of our discussion, and the abstract and the last paragraph of the introduction have also been amended accordingly.

Remarks from Reviewer #2

Schilder et al., present the results of phase I and IIa of the REGAIN clinical, a trial to restore hearing in individuals with long-standing, stable, sensorineural hearing loss (SNHL). Specifically, they use gamma secretase inhibitors (GSI) that inhibit the Notch pathway. The study was conducted as a multi-institutional clinical trial, using three different sites, and three different languages for the audiometric analyses. Results are presented for safety and toxicity as well as for efficacy. Study results are reported, and the authors conclude that 'GSI is safe and can lead to clinically relevant improvements in multiple measures of hearing function'. Careful review of the manuscript reveals significant concerns for bias in reporting study rationale and results. The reviewers do not agree with the presentation and summary of the data, discussion and conclusions. Below is a point-by-point summary of concerns.

Major comments:

- 1. The authors base the study on the potential of 'GSI to induce mammalian OHC regeneration and partial hearing restoration' which is also the first sentence of the abstract. Review of the manuscripts that were used as the rationale for the study reveals that there is no evidence to support efficacy of the proposed approach in adults with a stable hearing loss. The evidence in the literature indicates that GSI result in hair cell regeneration in P1 neonatal/cultured cochleae or in acutely deafened animals. The only three papers that are cited to support the use of this approach in adults include: (a) Hori et al – showed that only in acutely damaged guinea pig ears from ototoxic drugs, very few OHC can regenerate (and only in the non-sensory domain); (b) Mizutari et al – show that treatment of mice with GSI acutely after noise exposure can result in some improvement in the permanent threshold shifts in the low but not high frequencies. The same group reported in meetings that delayed treatment does not work; (c) Tona et al – show that guinea pigs acutely treated with GSI (within 7 days of noise exposure) had partial hearing restoration. Thus, there is no evidence in the literature to show that without acute injury in a mature inner ear – there would be any response to GSI treatment. Further, Hori et al show no response when the ototoxic treatment was incomplete. Finally, a paper from 2016 by the laboratory of Dr. Groves (27918591) shows that GSI inhibition results in no molecular response or changes in cochleae that are older than P6 in mice ('Mouse cochlear supporting cells become almost completely unresponsive to blockade of Notch signaling by six days of age'). One could argue that the authors wanted to try this approach despite the lack of proper support for it in the peer reviewed literature. In this case, the peer reviewed literature must be presented appropriately and the failure to reach statistically significant endpoints needs to be explained also in the context of the conflict between the study design (adults with stable hearing loss and no acoustic/ototoxic trauma) and hints for possible efficacy in the literature (neonatal ears or acutely injured ears in adults). In particular, the first sentence of the abstract should be changed to end with 'in the setting of acute injury only', the introduction should be modified to reflect these conflicts, and the discussion should be modified to address a rationale for a possible negative result of the study.*

Author response

The Mizutari 2013 paper reports on recovery of thresholds at the low frequency range after a single administration of the drug after acute noise exposures. Lack of responsiveness to Notch signaling blockade is apparent after the first postnatal week in mice in the absence of damage. Lack of response to blocked Notch signaling was seen in an undamaged cochlea in the cited paper and was not under conditions of hair cell loss where we have measured such a

response. Indeed, even when the cochlea was damaged in a similar study by the same authors (Maass et al, 2015), the noise level used was insufficient to cause hair cell loss. Others have also found a response after blocking Notch in adults: guinea pigs after exposure to noise and treated with siRNA against Hes1, a Notch downstream effector, showed threshold recoveries similar to ours (Du et al, 2018). The reviewer's points are reasonable with respect to long term deafness but matching of preclinical models to pathology and timing in human patients are difficult to achieve.

We have tried to include this in our amended introduction accordingly (page 7, line 84).

- 2. Partial reporting of adverse effects. The safety outcomes are reported only in a table without providing any detail on the most common adverse effects. Specifically, tinnitus is reported as experienced by 46% of the patients. However, no information is reported as for the duration of the tinnitus, whether the patients that reported tinnitus had tinnitus also before the study, whether the tinnitus was in the treated or control ear (specifically as the authors claim some improvement in hearing in the contralateral ear), and long term duration of tinnitus (according to IRB table 4, TFI is collected for all patients in weeks 6 and 12 of the study).*

Author response

We have added detailed information on adverse events as a supplementary table (table 2) This table includes all AEs reported as (possibly) related to IP as well as an overview of tinnitus reported as (possibly) related to IP. Out of 25 patient reporting tinnitus as an AE, 6 were ongoing at the end of the study; 20 were reported as mild and 5 as moderate. Patients could not always locate the side of their tinnitus; hence this information was not available for all related AEs. For TFI outcomes, please see the secondary outcomes in the Supplementary Information.

- 3. Reporting of improvement in PTA. The reporting of improvement in PTA revealed no change in the 2,4, and 8 kHz thresholds. It is not clear why the classic 500, 1000 and 2000 PTA are not also reported, specifically when animal studies suggested that efficacy, if present, is primarily in the lower frequencies. Furthermore – the PTA at these classic frequencies is the more important PTA for speech understanding. The authors proceed to report changes in hearing thresholds per individual frequencies, and find some improvement in 45%, 50% and 44% in one frequency at 3, 6 and 12 months, respectively. It is not clear if the improvement is in the same patient across tests. The data for the contralateral ear is not summarized in the context of individual patients, and fig 2 patient C shows improvement in the contralateral ear. The data should also be presented divided to frequencies 2kHz and lower (most relevant speech frequencies), and 4kHz and above (less relevant to speech understanding and most susceptible to test-retest variability from calibration or microphone placement), longitudinally per ear, and in comparison, to the other ear.*

Author response

We acknowledge the importance of accessibility of all trial data and therefore have compiled an overview of all pure tone audiometry data (treated and untreated ears) of all participants as Supplementary Information, Table 2. For reasons of readability, we have focused on the key pure tone audiometry data in the main manuscript text (results section) and have expanded Figure 2 to include pure tone audiograms of the treated ears of all participants over time.

To establish if an observed hearing improvement was consistent over time, we defined and analyzed this as a positive change of ≥ 10 dB(HL) at a certain frequency which was also present at two or more consecutive follow-up visits. The results of these analyses have been added to the relevant section (page 7, lines 143-148): see also our response to Reviewer 1).

- 4. In the section on outcomes for the untreated ear: pure tone improvements of 10 dB or greater for one or more frequencies are reported for 44-62% at the various time points. How many of these persons had improvements at a single frequency? How many had stable hearing at all frequencies? How many had hearing decline of 10 dB at one or more frequencies? How many had an improvement at some frequencies and a decline at other.*

Author response

We acknowledge the initial reporting of the results in the manuscript may have been incomplete. We have therefore expanded figure 2 to include all audiograms overtime of all participants (treated ear). In addition, we have added a table with all PTA data (treated and untreated ears) of all participants as Supplementary Information. We have also added the secondary efficacy outcomes analyses ≥ 10 dB in one or more frequencies in the treated ear, positive change of ≥ 10 in 2 or more adjacent frequencies, or ≥ 20 dB in a single frequency in the treated ear, and consistency over time defined as a positive change of ≥ 10 dB at a certain frequency in the treated ear which was also present at two or more consecutive follow-up visits) to the Methods (page 19, lines 438- 451) and the Results sections (see also response a to Reviewer 1).

- 5. There are several concerns about missing data that give the impression of selective reporting of positive findings. For example, the authors report trial site specific outcomes and limit the description of the pure tone findings to the German site whereas the comparable findings for the UK and Greek cohorts are not mentioned. Does this mean there was not a statistically significant improvement in pure tone thresholds for the subjects tested at the UK and Greek sites? Why is that not presented? Do the authors have an explanation for site differences?*

Author response

Indeed, there was no improvement in pure-tone thresholds for the UK and Greek participants. This information was included in the results section on trial site specific outcomes but was now moved to the supplementary appendix (page 6). The 24 patients at the UK site showed no significant change in mean pure-tone hearing threshold across 2, 4, and 8 kHz, nor in mean words-in-babble scores in the treated ear.

The 8 patients at the Greek site showed no significant change in mean pure-tone hearing threshold across 2, 4, and 8 kHz, nor in mean words-in-babble scores in the treated ear.

- 6. The authors do not provide enough detail about interpretation of the OAE data. Validity of the OAE must be taken into account using criteria such as those presented by Reavis et al. 2011 (PMID: 20625302) that consider a minimum level of noise, a minimum OAE amplitude, and a minimum SNR. When these criteria are not met, the OAE is considered absent. Given the degree of hearing loss required for study eligibility, it is expected that OAEs were absent at some or many frequencies. How was the absent OAE handled? Was*

this treated as missing data? Were the collected levels used without consideration of response validity? Was an arbitrary number assigned to the absent responses?

Author response

In this study we aim to investigate DPOAEs, beyond the typical criteria for SNR > 6dB and/or amplitudes >-20dB SPL which are used as thresholds for purposes of clinical diagnosis. Instead, considering the signal processing aspects of DPOAE measurements, in our analysis we include all available equipment-reported DPOAE amplitudes and SNR data. Additionally, in the current revision, along with the previous analyses, and to address reasonable concerns, we focus on DPOAE amplitudes rather than SNRs, and present additional results using estimates of the DPOAE amplitudes, which account for measurement and processing parameters such as the reported SNRs and number of signal averaging epochs (Dhar et al., 1998), (Backus, 2007). Moreover, along with the Wald-Wolfowitz Runs-test, we also present the results of additional tests for randomness of the DPOAE clinically meaningful patterns, based on indices of Bernoulli/Poisson statistics. These are included in the Results section (page 9, lines 146 – 152). A more detailed presentation of the analysis and results are presented in the Supplementary Information, which has been reorganized accordingly.

More specifically, in the part of the Supplementary Information for DPOAE results, in the beginning we provide a discourse on issues regarding the criteria used in clinical practice, and the rationale for our choice to use the raw DPOAE data together with the augmented analysis using SNR/averaging- based adapted DPOAE amplitudes. These analyses are presented in the following sections of the Supplementary Information.

Section 1 of the DPOAE Supplementary Information contains the analysis using the raw DPOAE data, as were given in the previous version of the manuscript. We also added the results of the Bernoulli-based test for pattern randomness to the Supplementary Information, regarding the analysis with the DPOAE amplitudes data, which take into account the SNR/averaging. The statistical analysis is the same to the analysis of the raw DPOAE data and also includes the Bernoulli-based test for pattern randomness. This section also references the methodology and processing for adapting the raw DPOAE amplitudes data to the SNR and the averaging during measurements' acquisition.

After Section 2, to facilitate comprehension, we also provide, a summarizing section with tables on our main findings in the previous Sections.

The results of the analyses using the estimates of DPOAE amplitudes as well as the additional tests for randomness are in congruence with the results that we reported in the initial version of the paper and support the detected indications of clinically meaningful effects.

7. OAE data shown in the supplemental tables show a number of frequencies where the SNR improved in the absence of a concurrent change in OAE amplitude itself at that same frequency. This implies that the noise levels were lower and suggests that these SNR changes do not represent improvement in cochlear function, but instead a reduction in environmental or physiologic noise during data collection.

Author response

Indeed, these are reasonable considerations. To resolve such ambiguities, we have included in the current revision an analysis regarding estimates of DPOAE amplitudes which also account for the SNRs. For details, please see our response to the previous comment.

8. *Do the authors have an explanation for the statistically significant effect of study site (“clinic”) on the OAE changes as shown in the supplemental tables.*

Author response

Regarding OAEs, we used the exact same measurement settings across all sites and in the UK and Germany the exact same hardware was used. This means we can largely exclude differences in the measurement methodology as an explanation for differences in this objective outcome.

9. *I struggle with the statistically based definition of a clinically relevant change in OAEs and recommend that the authors examine the literature on OAE stability and derive an evidence-based definition of a significant change in the OAE. This article, that provides guidance on serial monitoring of OAEs in clinical trials should be considered: Konrad-Martin D, et al, 2016, PMID: 27518137.*

Author response

Our analyses do not examine the data merely in terms of test-retest variability around a certain mean value of a number of measurements from a control population, but in terms of the expected mean (from many subjects) of the differences of such measurements from a specific value (clearly, in contrast to the distribution which is captured by test-retest variability of the varying measurements, these differences have a statistical distribution with an expected average value of 0 and higher variance). As the between-within subjects regression-based analyses (such as the one which we employ) offer estimates of the confidence intervals and inference of these means of differences from baseline and along time (again, from many subjects), then any such confidence intervals which do not include 0 shall be evident of a statistically significant difference (or change) from the reference value (the baseline in our case), regardless of the test-retest variability of the measurement values when observed one-by-one.

Although there is a significant amount of literature covering advances regarding the variability of observed DP responses (Konrad-Martin et al., 2016), and (Bader et al., 2021) cite several relevant works), at the best of our knowledge, in the literature there does not appear any relevant and universally adopted method which assesses the consistency or clinical importance of such differential DPOAE serial measurements from a baseline value, that span time periods of several months. For example, (Reavis et al., 2011) have introduced a heuristic approach for summary DPOAE metrics, however the assessment of trajectories over time of these metrics is performed using aggregate statistics (e.g. means, SDs, IRQs, etc.) together with PTA measures, in a framework of assessment of ototoxicity.

In this paper, based on the estimated confidence intervals for means of differences from baseline, and in a similarly heuristic approach, we consider “clinically-relevant” changes not on a purely statistical basis, but rather, on the basis of demand of consistency that these estimated confidence intervals exclude 0, for at least two consecutive instances in time (and not in an alternating fashion between positive and negative values), in order for the time course of changes from baseline to be considered as potentially meaningful.

10. *Discussion - the study, overall, had a negative result, and the discussion should be modified to reflect that and focus on it with a minor discussion of possible positive outcomes*

if the study would be heavily redesigned, rather than a focus on positive outcomes that are not founded in the presented data.

Author response

We have amended the discussion substantially and rephrased our conclusion. See Reviewer #1 comment and response c. We thank the reviewers for sharing their views.

Methods comments:

1. In a study that has the potential to inform therapeutic interventions for hearing loss, it is important that the methods section contain enough information to allow independent replication. The current description in the manuscript, the IRB document, and the other supplementary materials falls short. For example, were the audiologists blinded to previous hearing test results? If not, this is a potential bias. Were the earphones insert, supra-aural or circumaural? Did this vary between tests? What was the presentation level for the words in noise testing? Was care taken to ensure that the same word lists were not presented at each visit. What parameters were used to collect DPOAE data (number of sweeps, F1/F2 ratio?) What procedures were used to ensure good fit of the OAE probe and acceptable signal in the ear canal?

Author response

We have added substantial detail on study procedures to the methods section. Circumaural headphones were used for standard and extended high frequency Pure Tone Audiometry (page 16, lines 369 - 380). Blinding of the Audiologists to previous audiometric results was not included in the study design. Regarding the Words-in-Babble test, care was taken that the same word lists were not presented at each visit, the settings for speech in noise testing have been added (pages 17-18, lines 382 - 409).

With regards to the parameters used to record DPOAEs, we used an F2/F1 Ratio of 1.22, 8 Bands per octave, 3 Blocks, 90 sweeps, and 5 retries, for both low-level (65/55 dB SPL) and high-level (70/70 dB SPL) tone primaries. At the beginning and at the end of each measurement, the device software checked the correct fit of the earpiece and the probe in the ear according to the manufacturer's recommendation (page 18, lines 410 – 415).

2. The initial audiogram, which was obtained 0-14 days prior to the intervention, was used to determine eligibility, and serve as baseline. Stable hearing prior to treatment is important consideration for subject participation in a study aiming to show an effect of intervention, and this is one of the exclusionary criteria for subject enrolment in this trial (IRB supplement). However, there is no mention of methods used to ensure hearing stability and no replication of the pre-treatment audiogram. This is a shortcoming.

Author response

We appreciate this concern and have added the following section to the discussion: "We determined pre-trial stability of patients' hearing loss upon based upon their medical history and review of their previous audiograms. Eligibility was confirmed by a screening audiogram within 4 weeks of the first LY3056480 injection. One may argue that a pre-trial lead-in time with multiple audiological assessments would have ascertained stability of hearing loss of our participants" (see page 11, lines 219 - 223).

3. There are several criteria used to define a significant change in pure tone thresholds. This includes the widely used and accepted ASHA (1994) definition that requires a 10 dB change at two consecutive frequencies, or a 20 dB change at a single frequency. The authors of the current manuscript define a significant change in hearing based on the pure tone average of 2, 4 and 8 kHz as an improvement ≥ 10 dB and, mathematically, this would achieve the ASHA criteria. However, the authors subsequently report improvement in hearing by 10 dB at one or more frequencies. An improvement in hearing at a single frequency by 10 dB is not acceptable evidence and it is recommended that the authors re-evaluate their data using criteria such as those established by ASHA.

Author response

We thank the reviewer for this important comment. Please see also Reviewer #1 comment a. and our response. This comment prompted us to conduct an additional analysis of pure-tone thresholds which we have added to the manuscript Methods and Results.

4. What is the basis for the 2dB change in the speech in noise SNR as the cut point? Provide references and a rationale based on the known psychometric functions of each test used to evaluate speech in noise listening.

There is no guidance in the literature nor from regulatory bodies regarding the level of SNR to be used in trials of hearing regeneration. Hence, we chose a 2 dB level based on relevant literature based on the WIB test. Dillon (2012) reported that for every 10 dB of hearing loss, the subject requires an additional 1–3 dB of SNR to maintain their unaided intelligibility. This reference was added (page 19, line 456).

Regarding sentence-based tests, Killion (2002) reported for the QuickSIN that adults with normal hearing needed an average SNR-50 of 2 dB to repeat 50 percent of the words correctly and Wardenga (2015) described for hearing impaired subjects up to 46.7 dB (PTA4) a SNR SD of 1.17 dB (International Matrix Test). As two times SD gives the 95%-confidence interval for gaussian distributions, a 2 dB shift compared to normal values is a reasonable borderline to suspect a significant SIN shift in our study.

Minor comments

1. The authors address hearing loss as a single disease and group patients with different durations of hearing loss, different severities of hearing loss, and different audiogram configurations as one group. This type of approach assumed a 'one size fits all' for a treatment that strongly decreases those chances of identifying successful outcomes given the heterogenous nature of hearing loss.

Author response

Developments within the field of novel hearing therapeutics over the past 6 years, have brought this to the spotlight (McAlpine 2022). We have addressed this in our revised discussion (pages 11 - 12, lines 211 - 218): " Our trial has generated important learnings about the design and delivery of early phase trials of novel hearing approaches. Through a detailed medical history and audiological assessment at the screening visit, checked against our in- and exclusion

criteria, we aimed to include patients with SNHL most likely due to outer hair cell loss. We acknowledge that current tests of auditory function do not allow for deep phenotyping and therefore heterogeneity of our patient population in terms of underlying molecular mechanisms of SNHL may have diluted the effects of our highly targeted treatment. Collaborative efforts towards understanding the deep geno-phenotype of hearing and hearing loss are urgently needed.

We have also provided mean pure tone levels in the treated ear in the new figure 2 (page 7) and individual PTA hearing levels of all patients over the course of the trial in the new Figure 3 (page 8).

- 2. Discussion – the authors explain that hearing was measured up to 16 kHz as higher frequencies are important for understanding speech in noise. 4 and 6 kHz are important for speech in noise, but the reviewers are not familiar with literature showing that 12-16 kHz (which are more variable) – have any effect on speech in noise. Furthermore, for none of the patients had the hearing handicap improved.*

Author response

At the time we designed our trial we decided to measure extended high frequency PTA because of our consideration that this was relevant for SiN performance (Hunter , 2020). We have added to the manuscript the corresponding literature (page 10, line 200).

- 3. Discussion – ‘going forward we would recommend future trials to assess pure-tone hearing thresholds as 1-2 dB(HL) steps... matrix test’. Please remove this sentence. There are no data to support this claim in the results. Furthermore, there are no data to show clinical significance of 1dB. Assessing hearing at steps that are clinically insignificant and are at a significant risk of test-retest variability will muddy the literature and result in irresponsible use of resources.*

Author response

We have removed this sentence.

- 4. ‘our findings support our hypothesis and suggest DPOAEs can be used to detect HC regeneration’ – please remove this sentence. There results do not support this claim.*

Author response

We have removed the sentence in the manuscript.

- 5. ‘next steps could include testing of efficacy in higher dosages’ – the presented data do not support this suggestion. Perhaps restructuring the study to address instances of acute hearing loss would be more appropriate and grouping patients by type and duration of hearing loss. However, the current study does not support efficacy.*

Author response

In future trials we will address the highest unmet clinical need, including acute hearing loss.

6. *Discussion – closing sentence – please modify as conclusion is not supported by data.*

Author response

We have amended the discussion substantially and rephrased our conclusion. See Reviewer #1 comment and response c. We thank the reviewers for sharing their views.

7. *Discussion – improvements in the contralateral ear: this discussion should be completely removed. There is no quantification or rigorous study of hearing changes in the contralateral ear. The proposed mechanisms by the group are highly unlikely. And if this remains, error and issues with test-retest reliability at the frequencies that the authors do detect improvement should be discussed and considered as well.*

Author response

We have removed this section from the discussion.

Remarks from Reviewer #3:

This is a proof-of-concept in a Phase I/IIa multiple-ascending-dose open-label trial to establish the safety and efficacy of a regenerative drug treatment for sensorineural hearing loss. Well written paper and well design study. The primary endpoints were pure-tone audiometry at 3 months. There is no serious concern on the safety and tolerability, but improvement was not observed and maximum tolerated dose was not met. I have only comments on the clarity of the design:

- a. *CONSORT: in phase 1, could you please check the consistency as 12 patients were not enrolled but justification were given for 13 patients*
- b. *A sample of 15 is quite small for a 3+3 design, could the authors clarify the design and the choice of 250 microgram, what was the first dose 25 or 125 or 200 or 250 microgram? A diagram of safety results by cohorts will be more informative than a table.*
- c. *Safety and tolerability endpoints in the SAP (6.2.3 paragraph) are 6 endpoints, these need to be all reported in a supplement table as planned.*
- d. *The primary outcome was well investigated, but the secondaries need to be reported as stated in the SAP for consistencies.*
- e. *Discussion: please elaborate on the dosages and tolerability of higher dose and the progression to phase IIa, given 250 microgram seems safe but maybe not MTD. For example, would you re-do the phase 1 study with higher doses?*

Author response

- a. We apologise for this error from our side, we have corrected the CONSORT diagram (Figure 1, Page 5) accordingly.
- b. In our phase I trial the patients received in ascending dose cohorts 25µg, 125µg, 200µg, and 250µg applied in 0.5ml (see Results, page 5, lines 92 - 98). We have added a section about our 3+3 design and choice of dosage to the discussion (page 11, lines 224 - 229). As the number of patients in the phase I trial are very small (n=3) we do not believe a by cohort diagram of safety results will add much.
- c. We have added a document providing all safety and tolerability data (including the secondary measures) to the Supplementary Information.

- d. See our response to Reviewer #3 comment c.
- e. See our response to Reviewer #3 comment b. With the current formulation, we cannot reach a higher dose due to solubility of the Investigational Product. A new trial would involve developing a new formulation or, in the light of good to tolerability and safety shown in this trial, administer more injections of the current formulation.

Remarks from Reviewer #4

The paper by Schilder et al. written on behalf of the REGAIN consortium reports on safety and efficacy of LY3056480 in patients with sensorineural hearing loss (SNHL) in phases I and IIa studies. The authors claim to have “delivered proof-of-concept in a Phase I/IIa multiple-ascending-dose open-label trial”. However, the authors also confess that “the primary endpoint of >10 dB (HL) improvement across 2, 4, 8 kHz was not met”. This means that the REGAIN study did not deliver proof-of-concept as defined in the study protocol and ought to be considered negative. The careful language describing the findings is critical because humans suffering from hearing loss are expecting positive news from clinical research on means expected to help them.

Author response

Please see our responses to the relevant comments of Reviewer #1 and #2. We have amended the discussion substantially and rephrased our conclusion. We thank the reviewers for sharing their views.

Other critical issues which were not addressed adequately by the authors are:

- 1. German center delivered ample improvement of audiologic biomarkers whereas both the British and Greek centers did not at all.*
- 2. Safety and tolerability of the LY3056480 may be confirmed by the phase I study. However, there is little attention placed on administration procedures. That is critical for multiple drug administration and compliance. It has not been critically discussed and no alternatives have been suggested.*
- 3. In addition, a clear-cut definition of adverse events that suggests toxicity from the investigated drug (DLT, dose-limiting toxicity) is missing. It is therefore unclear, if the targeted probability of the 3+3 for a DLT is appropriate. It is unfortunate that doses higher than 250 µg were not considered in the phase I trial.*
- 4. The authors state that higher efficacy could be achieved by elevating the dose of LY3056480, however the maximum dose tested in the phase I was 250 µg. This dose cannot be increased due to regulations unless another phase I study is performed. In other words, increasing the dose in order to increase efficacy based on the results of the current phase I study should not be suggested. The only way to increase the dose is by increasing the number of injections at a maximum tested dose of 250 µg/injection over time.*

Author response

1. Please see our responses to Reviewer #2 major comment 5. We have discussed the observed differences in outcomes across sites in our Consortium and have addressed this in the discussion as follows (page 12, lines 245 – 246) “Whether a smaller step size,

improves accuracy (Jerlvall, 1986) and therefore detection of efficacy signals and may explain the differences across trial sites remains open for debate”.

2. See our response to Reviewer #2 Methods comment 2. We have added substantial detail on trial procedures to the methods section, including drug administration (page 15, lines 323 – 332). Regarding alternatives for drug administration, we have added the following to the discussion (page 11, lines 233 - 236): “At the time of development of the trial, we discussed various options for drug administration among our Consortium and with our UK patient panel. At that time, patients shared a strong preference for intratympanic injections in an outpatient setting over a surgical approach that may require a general anaesthetic to deliver the drug directly onto the round window”. Please see our response to comments b and e of Reviewer #3. With the maximum dose of 250 µg (due to solubility of the formulation and middle ear volume) we did not reach Dose Limiting Toxicity.
3. We agree with this suggestion. See our response to comments b and e of Reviewer #3. With the current formulation, we cannot reach a higher dose due to solubility of the IP. A new trial would involve developing a new formulation or, in the light of good to tolerability and safety shown in this trial, administer more injections of the current formulation.

This study will profit from applying objective interpretation of the data as regulatory agencies require. The report, as stands, is delivering a message of success. However, the data do not support that message, particularly not in case of efficacy.

Author response

Please see our responses to the relevant comments of Reviewer #1 and #2. We have amended the discussion substantially and rephrased our conclusion.

The REGAIN trial is leading to the follow up trial named VESTA. VESTA is a double blinded, randomized, placebo controlled, multi-center efficacy phase II trial comparing three different dosing regimens of LY3056480 based on the maximum single dose of 250 µg explored in the REGAIN trial. Sufferers from mild to moderate SNHL will be recruited. Four injections of 250 µg of LY3056480 or placebo will be administered transtympanically into one ear. The primary outcome measure of efficacy of local treatment with LY3056480 in terms of hearing after 6 months will be the number of responders to therapy with at least a 2 dB improvement in an adaptive manner in noise test compared to placebo. All this is a result of experience gained from the REGAIN trial, but these conclusions are not explained in the current report.

Author response

The follow-up trial VESTA was registered in the US on clinicaltrials.gov in preparation for IND submission to FDA. Due to a lack of resources, there are no current plans to conduct this trial, hence registration is currently on hold.

There is no criticism regarding quality of study protocol, procedures employed, or statistical evaluation of the data. However, the 3+3 design might be a little old-fashioned, but might be the easiest to implement under complicated study conditions. In addition, since the study does not have control group, it is not randomized. This may be seen as limitation, since in pre/post comparisons, the observed effect may be considered as higher.

Author response

Please see our response to comment b by Reviewer #3. We have added a section about the choice of our study design to the discussion (page 11, line 224 – 232). We chose a traditional 3+3 dose escalation study design, starting with the ‘no observed adverse effect level’ dose from our pre-clinical studies and ending with the maximum dose based upon solubility of the formulation and middle ear volume, over an innovative Phase I design. This is because at the time our trial was initiated, there were no data on the safety of novel hearing therapeutics. For future trials one may agree with regulators on more rapid dose escalation or integrated Phase I/II designs. For the same reason, we chose to inject one ear only. Now that more data are available on the safety of hearing regenerative and considering the observed outcomes in the untreated ear, we would recommend future trials adopt a placebo (diluent or saline) -controlled design, using patient-level randomization.

References

- Mizutari, K, et al. Notch inhibition induces cochlear hair cell regeneration and recovery of hearing after acoustic trauma. *Neuron* 77, 58-69 (2013).
- Maass JC, Gu R, Basch ML, Waldhaus J, Lopez EM, Xia A, Oghalai JS, Heller S, Groves AK. Changes in the regulation of the Notch signaling pathway are temporally correlated with regenerative failure in the mouse cochlea. *Front Cell Neurosci.* (2015) Mar 31;9:110. doi: 10.3389/fncel.2015.00110. PMID: 25873862; PMCID: PMC4379755.
- Du X, Cai Q, West MB, Youm I, Huang X, Li W, Cheng W, Nakmali D, Ewert DL, Kopke RD Regeneration of Cochlear Hair Cells and Hearing Recovery through Hes1 Modulation with siRNA Nanoparticles in Adult Guinea Pigs..*Mol Ther.* (2018) y 2;26(5):1313-1326. doi: 10.1016/j.ymthe.2018.03.004. Epub 2018 Mar 10.PMID: 29680697
- Backus, B. C. (2007). Bias due to noise in otoacoustic emission measurements. *The Journal of the Acoustical Society of America*, 121(3), 1588–1603. <https://doi.org/10.1121/1.2434831>
- Bader, K., Dierkes, L., Braun, L. H., Gummer, A. W., Dalhoff, E., & Zelle, D. (2021). Test-retest reliability of distortion-product thresholds compared to behavioral auditory thresholds. *Hearing Research*, 406, 108232. <https://doi.org/10.1016/j.heares.2021.108232>
- Dhar, S., Long, G. R., & Culpepper, N. B. (1998). The Dependence of the Distortion Product $2f_1 - f_2$ on Primary Levels in Non-Impaired Human Ears. *Journal of Speech, Language, and Hearing Research*, 41(6), 1307–1318. <https://doi.org/10.1044/jslhr.4106.1307>
- Konrad-Martin, D., Poling, G. L., Dreisbach, L. E., Reavis, K. M., McMillan, G. P., Lapsley Miller, J. A., & Marshall, L. (2016). Serial Monitoring of Otoacoustic Emissions in Clinical Trials. *Otology & Neurotology*, 37(8), e286. <https://doi.org/10.1097/MAO.0000000000001134>
- Reavis, K. M., McMillan, G., Austin, D., Gallun, F., Fausti, S. A., Gordon, J. S., Helt, W. J., & Konrad-Martin, D. (2011). Distortion-Product Otoacoustic Emission Test Performance for Ototoxicity Monitoring. *Ear & Hearing*, 32(1), 61–74. <https://doi.org/10.1097/AUD.0b013e3181e8b6a7>
- Killion MC, Niquette PA, Gudmundsen GI, Revit LJ, Banerjee S. Development of a quick speech-in-noise test for measuring signal-to-noise ratio loss in normal-hearing and hearing-impaired listeners. *J Acoust Soc Am* 116, 2395-2405 (2004).

- Wardenga N, Batsoulis C, Wagener KC, Brand T, Lenarz T, Maier H. Do you hear the noise? The German matrix sentence test with a fixed noise level in subjects with normal hearing and hearing impairment. *Int J Audiol* **54 Suppl 2**, 71-79 (2015).
- McAlpine D, Goldman, D., Schilder, AGM., ENT & Audiology News 2022, September/October, Volume 31 No 4 page 33-34. Mind the gap – developing a sustainable pipeline for hearing therapeutics (2022). <https://www.entandaudiologynews.com/features/audiology-features/post/mind-the-gap-developing-a-sustainable-pipeline-for-hearing-therapeutics>

REVIEWERS' COMMENTS

Reviewer #2 (Remarks to the Author):

The authors have made commendable improvements to the data presentation in the manuscript, enabling readers to gain a clearer understanding of the trial results. Many of the reviewer comments, however, were discussed but not fully addressed.

Reviewer #4 (Remarks to the Author):

The authors responded to all my criticism and removed enthusiastic and not supported by data statements from the manuscript. The overall value of the manuscript has therefore increased. In its present form the work reports important novel clinical findings and discusses them critically. The overall manner of data presentation has also been radically improved and follows now principle of objective presentation of clinically collected data on humans. Current discussion is appropriate and does not contain any unnecessary emotional statements.

Reviewer #5 (Remarks to the Author):

Authors have adequately addressed Reviewer #3's previously raised concerns